# Fatal Neonatal Sepsis Associated with Human Adenovirus Type 56 Infection: Genomic Analysis of Three Recent Cases Detected in the United States

**DOI:** 10.3390/v13061105

**Published:** 2021-06-09

**Authors:** William R. Otto, Daryl M. Lamson, Gabriel Gonzalez, Geoffrey A. Weinberg, Nicole D. Pecora, Brian T. Fisher, Kirsten St. George, Adriana E. Kajon

**Affiliations:** 1Division of Infectious Diseases, Department of Pediatrics, The Children’s Hospital of Philadelphia (CHOP), Philadelphia, PA 19104, USA; OTTOW@chop.edu (W.R.O.); FISHERBRIA@chop.edu (B.T.F.); 2Center for Pediatric Clinical Effectiveness, CHOP, Philadelphia, PA 19104, USA; 3Wadsworth Center, New York State Department of Health (WC-NYSDOH), Albany, NY 12201, USA; daryl.lamson@health.ny.gov (D.M.L.); kirsten.st.george@health.ny.gov (K.S.G.); 4National Virus Reference Laboratory (NVRL), University College Dublin, D04 V1W8 Dublin, Ireland; gabo.gonzalez@ucd.ie; 5International Collaboration Unit, Research Center for Zoonosis Control, Hokkaido University, Sapporo 001-0020, Japan; 6Department of Pediatrics, University of Rochester School of Medicine & Dentistry Rochester (UoR), Rochester, NY 14642, USA; Geoff_Weinberg@URMC.Rochester.edu; 7Department of Pathology and Laboratory Medicine, UoR, Rochester, NY 14642, USA; Nicole_Pecora@URMC.Rochester.edu; 8Immune Dysregulation Program, CHOP, Philadelphia, PA 19104, USA; 9Department of Biomedical Science, University at Albany, SUNY, Albany, NY 12222, USA; 10Lovelace Biomedical Research Institute, Albuquerque, NM 87108, USA

**Keywords:** adenovirus, neonatal sepsis, genomics, intermediate strain, species HAdV-D

## Abstract

Background: Human adenovirus (HAdV)-D56 was first described in 2011 by genomics analysis of a strain isolated in France in 2008 from a fatal case of neonatal infection. Since then, it has been reported in cases of keratoconjunctivitis and male urethritis. Three epidemiologically unrelated fatal cases of neonatal sepsis associated with infection by HAdV-D strains with a similar genetic makeup were documented in the United States between 2014 and 2020. Methods: Whole genome sequences were obtained for the isolated strains, and genomics analyses were conducted to compare them to phylogenetically related HAdV-D genomic sequences available in GenBank. Results: The three new US strains were indistinguishable by in silico restriction enzyme analysis. Their genome sequences were 99.9% identical to one another and to the prototype strain isolated in 2008 from a similar context of disease. The phylogenetic reconstruction revealed a highly supported clustering of all HAdV-D56 strains isolated in various countries since 1982. Our comparison to serologically intermediate strains 15/H9 described in the literature indicated that HAdV-D56-like viruses have circulated worldwide since the late 1950s. Conclusion: As with other HAdV-D genotypes with the ability to infect ocular and genital mucosae, the risk of severe prenatal or perinatal HAdV-D56 infection must be considered.

## 1. Introduction

Neonatal disease associated with human adenovirus (HAdV) infection, although relatively rarely reported, is frequently disseminated and can be fatal [1,2]. The types classified within species B, C, and D have been the most frequently detected among the published cases for which virus typing was conducted [3,4].

Since the description of serotype 51 by De Jong et al. [5] in 1999, more than 40 new HAdV-D genotypes have been described based on the bioinformatics analysis of whole genome sequences, revealing novel intertypic genetic recombinants or novel penton base, hexon, or fiber gene sequences. A current list of recognized types can be found at http://hadvwg.gmu.edu. Human adenovirus-56 was first described in 2011 as an emergent pathogen by Robinson et al. [6] via genomics analysis of a respiratory isolate obtained from a 2008 fatal case of neonatal infection in France [7]. This virus was originally described as having a type 15/29-like hexon gene, by amplification and sequencing of the hexon hypervariable regions 1–6, and a serotype 9-like fiber by hemagglutination inhibition. Importantly, its transmission to unprotected healthcare staff was documented to cause keratoconjunctivitis [7]. Since 2011, and under a new designation, HAdV-56, this virus has been reported in cases of epidemic keratoconjunctivitis (EKC) in Japan [8] and in an outbreak in China [9] as well as in cases of male urethritis with concurrent conjunctivitis in Japan and Spain [10,11,12].

In this paper, we report the comprehensive genetic characterization of three new strains of HAdV-56 isolated from epidemiologically unrelated fatal cases of neonatal sepsis detected in the United States between 2014 and 2020. We also report the results of a comparative analysis that allowed us to demonstrate that viruses with very similar, if not identical, characteristics have circulated in various continents since the late 1950s.

## 2. Materials and Methods

### 2.1. Case Identification and Clinical Data Abstraction 

Cases were retrospectively diagnosed and originally investigated based on the clinical interest they elicited at Nationwide Children’s Hospital, the University of Rochester (UoR), and the Children’s Hospital of Philadelphia (CHOP), where medical records were retrospectively reviewed.

Case 1 (Ohio, 2014). An 8-day old, 23-week gestational age, premature female infant developed sepsis, pneumonia, hepatitis, and thrombocytopenia. Human adenovirus DNA was detected in the nasopharynx and blood prompting therapy with cidofovir and intravenous immunoglobulin (IVIG). Despite these interventions, the infant died at 18 days of age. Detailed clinical information for this case was previously reported by Moallem et al. [13]. Human adenovirus DNA was detected in the nasopharynx and blood. The HAdV strain isolated from a nasopharyngeal aspirate, NCH-PRO95, was originally partially characterized at Lovelace Biomedical Research Institute (LBRI) and typed as an intertypic recombinant HAdV-D with a type 15/29 hexon gene (H) and a type 9 fiber gene (F) as reported [13];Case 2 (New York, 2019). A 9-day old, full-term male infant experienced fever and poor feeding and was admitted to a local hospital in NY state. The infant was well for the first week of life, and his New York State Newborn Screening was negative for all diseases tested for in the standard screening program at the state newborn screen laboratory (https://www.wadsworth.org/programs/newborn/screening/screened-disorders). Physical examination and initial laboratory studies were nondiagnostic, and intravenous ampicillin, gentamicin, and acyclovir were administered. Bacterial cultures of blood and urine specimens were negative; surface cultures and blood PCR for herpes simplex virus were negative as well. A PCR-based respiratory virus panel test (Panther Fusion, Hologic, Inc., Marlborough, MA, USA) conducted on a nasopharyngeal specimen was positive for both adenovirus and rhinovirus. Because of progressive hypoxia and continued fever, the patient was transferred to a children’s hospital after 72 h of therapy. Further history was obtained that the infant’s mother and grandmother had recent episodes of conjunctivitis around the time of the child’s birth. The physical examination continued to be unremarkable except for temperatures up to 40.2 °C. A chest radiograph showed bilateral diffuse opacities indicative of either infiltrates or atelectasis. Over the subsequent 48 h, he developed worsening tachypnea and oxygen desaturation and was transferred to the pediatric intensive care unit (ICU). Bacterial cultures of blood and urine continued to be negative. His respiratory function rapidly declined, and intubation with mechanical ventilation, followed by provision of inhaled nitric oxide and later extracorporeal membrane oxygenation (ECMO) were required. Serum PCR assays conducted at Eurofins Viracor-IBT laboratories (Lee’s Summit, MO, USA) for enterovirus were negative, but those for HAdV were positive at 1.6 × 10^9^ genome copies/mL. Cidofovir was administered intravenously and ECMO maintained, but the infant had continued respiratory failure and developed hepatorenal failure and coagulopathy. He died at 19 days of age;Case 3 (Pennsylvania, 2020). A 14-day old, full-term infant male presented with a 5-day history of poor feeding and new-onset respiratory distress and lethargy. He previously had been noted to have bilateral conjunctival erythema at 6 days of life. Due to the severity of his illness, he was taken to an outside emergency department, where he was noted to have severe retractions and multiple apneic events. He was intubated but continued to have hypoxemia despite escalation to high-frequency oscillatory ventilation, so he was transferred to a children’s hospital where he was cannulated onto ECMO. He required significant vasopressor support and also had renal failure requiring renal replacement therapy. Cerebrospinal fluid studies were unremarkable. He was given broad spectrum antimicrobials, including vancomycin, cefepime, and acyclovir. Bacterial cultures were negative. An in-house respiratory virus quantitative real-time PCR panel [14] was positive for HAdV. A serum HAdV PCR also tested positive. Due to the severity of his presentation, IVIG was given daily for 2 days. Cidofovir was held due to the concerns for renal toxicity. Because he continued to have clinical instability and repeat serum HAdV PCR testing 3 days later was positive, he was given cidofovir dosed at 5 mg/kg with hyperhydration. Over the following weeks, serial serum HAdV PCR testing remained positive, and he received 2 additional doses of cidofovir on a weekly basis. He also received 5 additional doses of IVIG. Despite these interventions, there was little change in his clinical status, and he continued to require ECMO support due to the fact of respiratory failure and severe hemodynamic instability. On day 42 of life, he was noted to have worsening leukopenia and thrombocytopenia with associated hemodynamic instability. He then developed both pulmonary and gastrointestinal hemorrhage. Due to the bleeding, he was decannulated from ECMO and died on day 43 of life.

### 2.2. Virus Isolation and Initial Molecular Typing

All HAdV-positive clinical specimens were inoculated into conventional virus culture tubes of A549 cells (American Type Culture Collection, Manassas, VA, USA, ATCC CCL-185) for virus isolation and initial propagation. Infected monolayers were monitored for development of cytopathic effect (CPE) for one week and harvested when extensive CPE was observed. Virus isolates were propagated in 75 cm^2^ flasks for purification of intracellular viral DNA as previously described [14]. Molecular typing was initially carried out by PCR amplification and Sanger sequencing of hypervariable regions (HVRs) 1–6 of the hexon gene and the complete fiber gene followed by BLAST (https://blast.ncbi.nlm.nih.gov/Blast.cgi) analysis [13,14]. Molecular type identities were assigned based on the identity of the closest match.

### 2.3. Whole Genome Sequencing and Annotation

Purified genomic viral DNA was prepared from the three isolates for whole genome sequencing in an Illumina MiSeq instrument at the NY State Department of Health (NYSDOH) Wadsworth Center as previously described [15]. Paired fastq sequence files for each sample were imported into Geneious Prime version 2020.2.4 (Biomatters, Ltd., Auckland, New Zealand), trimmed using BBDuk, and error corrected and normalized using BBNorm, before alignment to the reference genome HM770721. The consensus sequences were further annotated using this genome as a reference in VAPiD [16] and uploaded to GenBank using the NCBI Bankit tool. Uploaded sequences are available under accession numbers MW805358, MW805359, and MW805360.

### 2.4. Genomic Analyses

The genomic sequences of the three newly identified strains of HAdV-56 were compared to publicly available whole genome sequences in GenBank within the *Adenoviridae* family (taxid: 10508) using the online BLASTn program [17] with parameters e-value < 0.01 and word size 7. Identified HAdV genomic sequences with similarity > 98% and a selection of reference genomic sequences were aligned using MAFFT with the FFT-NS-I [18]. The analysis included the WGS for the first case of HAdV-D56 infection reported by Henquell et al. (GenBank #HM770721), various complete genomic sequences available from GenBank for virus strains designated as HAdV-D56 or exhibiting similar genetic identities for the penton base (P), hexon (H), and fiber (F) genes (Table 1), and the genomic sequences for the prototype strains of the phylogenetically related genotypes HAdV-D9 (GenBank #AJ854486), -D10 (GenBank #AB724351), -D15 (GenBank #AB562586), -D26 (GenBank #EF153474), -D29 (GenBank #JN226754), -D88 (GenBank #MF476842), and -D94 (GenBank #KF268201).

A maximum likelihood phylogenetic tree to explore the relation among sequences was inferred using RAXML-GUI v2.0 [19] with the GTRGAMMA as a substitution model and 1000 repetitions to calculate the bootstrap branch support.

In silico restriction enzyme analysis with endonucleases Bam HI, Bgl II, Bst EII, Hind III, Pst I, Sac I, Sal I, and Sma I was carried out in Geneious Prime version 2020.2.4 (Biomatters, Ltd., Auckland, New Zealand).

For a more detailed exploration of genomic sequence similarities, a sliding window analysis was performed with Simplot V3.5.1 [20] using a 500-nucleotide sliding window, a 50-nucleotide step size, using GapStrip, on a Kimura distance model, and Ts/Tv = 2.0. The consensus sequence for the three strains characterized in this study was used as the query in the comparison with the genomes of HAdV-D9, -D10, -D15, -D26, -D29, -D88, and -D94 listed above. A pairwise analysis of sequence similarities across the entire panel of genotypes and strains was carried out in MEGA7 v7.0.25 [21]. The similarity between pairs of compared sequences was expressed as a percentage, and a heat map graphical representation of the data matrix was built using R version 4.0.5 [22].

In order to parse amino acid differences for the polypeptides encoded in selected regions of the genome among the different HAdV-D types, amino acid sequences were examined with the method of proteotyping adapted from Obenauer et al. [23]. The concatenated protein sequences of penton base, hexon, E3 proteins, and fiber were analyzed. Amino acid signature patterns were derived from the maximum likelihood phylogenetic tree-guided sequence alignment, indicating amino acid sites with polymorphisms relative to the most frequently occurring residues with a frequency-based color coding.

## 3. Results

### 3.1. Preliminary Virus Typing

Virus isolates were only recovered from respiratory specimens. As originally reported for Case 1 [13], the initial molecular characterization of the HAdV isolated from the nasopharyngeal specimen of case 2 at NYSDOH (strain IDR1900044114) and of the HAdV isolated from the nasopharyngeal specimen of case 3 at LBRI (strain VIR209329), identified a HAdV-D with a HAdV-D15/29-like hexon gene and HAdV-D9-like fiber gene indicating them to be candidate HAdV-D56 strains.

### 3.2. Genetic Analysis of Virus Isolates

The complete genomic sequences of the three US strains isolated from epidemiologically unrelated fatal cases of neonatal sepsis described above were obtained with an average coverage of 28X for strain D56-XII, 17X for strain D56-XV, and 31X for strain D56-XVI. The genomic sequences were 99.9% identical to one another, and 99.9% identical to the prototype strain isolated in 2008 in France from a similar context of disease [7] and to the earliest clinical isolate available for comparison, D56-I from Germany, 1982. The maximum likelihood phylogenetic reconstruction revealed a highly supported clustering of all examined HAdV-D56 strains, with other related genotypes segregating in distinguishable clades (Figure 1). The HAdV-D56 cluster has a sister clade comprising genotypes: D9, -D88, and -D94 with overall sequence similarities around 98.5% (Figure 2). The in silico restriction enzyme analysis (Figure 3) confirmed the close relationship among the four genotypes and portrayed the examined HAdV-D56 strains as a group of closely related genomic variants with identical Bgl II and Hind III digestion profiles. Fifteen of the 16 examined strains also shared the same prototype-like Bam HI profiles.

The three US strains were indistinguishable from one another by in silico REA with eight endonucleases, and, as shown in Table 2, only 14 non-synonymous point mutations were identified in these genomes compared to the 2008 French strain.

A sliding window analysis along the genome was used to examine the similarity among the closely related genotypes HAdV-D9, -D10, -D15, -D26, -D29, -D56, -D88, and -D94. As shown in Figure 4, the divergence between HAdV-D56 and the other types concentrated in the loci encoding the major capsid proteins penton base, hexon, and fiber, and also in the E1, E4, and E3 transcriptional units with particularly high divergence in windows located in the E3-CR1β coding region. As previously reported [6,24], HAdV-D56 showed high similarity in the penton base, hexon, and fiber to types -D9, -D15, and -D9, respectively. As expected, differences to the closely related H15 F9 genotypes -D88 and -D94 were evident in the penton base coding region, accounting for the separate clustering in the phylogenetic tree.

The phenotypic implications of the genetic diversity detected among the selected panel of HAdV-D viruses were examined by proteotyping of the predicted sequences for penton base, hexon, E3, and fiber proteins. As shown in Figure 5, the different profiles confirmed the similarity of HAdV-D56 strains to genotypes -D9, -D10, and -D26 in the penton base; the similarity to genotypes -D15, -D29, -D88, and -D94 in the hexon protein and the similarity to genotypes -D9, -D88, and -D94 in the fiber. Overall, the HAdV-D56 E3 cassette of proteins resembled more closely those of genotypes D9, D88, and D94 with two readily distinguishable variants of CR1β and CR1γ present in the genomes of all the three new HAdV-D56 strains and in those of all Japanese strains isolated since 2010.

## 4. Discussion

The epidemiology of neonatal adenovirus infection is still poorly characterized and the mechanisms of infection unclear but probably multiple. Vertical transmission during vaginal delivery is a very plausible scenario in view of the various reports of detection of HAdV in the female genital tract [25,26,27]. For some cases, strong evidence of prenatal ascending intrauterine infection or transplacental transfer has been reported [28,29,30].

The possible sexual transmission of infection of various HAdV types, and HAdV-D types in particular, has been considered and merits further investigation given the number of reported cases of associated male urethritis and cervicitis [25,26,31,32,33,34]. In this context of disease, HAdV-D56, an intertypic recombinant genotype (P9H15F9) described in 2011 [6], has recently attracted a lot of attention as a result of its detection in cases of male urethritis with concurrent conjunctivitis and in cases of epidemic keratoconjunctivitis [9,10,11]. Like the prototype strain isolated in 2008 [7], the three strains characterized in this study were isolated from cases of bacterial culture-negative neonatal sepsis, documenting the potential for this viral infection to be fatal in the neonatal period. Notably, in case 2, a history of recent conjunctivitis had been recorded for the patient’s mother and grandmother, indicating that horizontal transmission may play a role in some cases.

Our genomics analysis included all available whole genome sequences in GenBank identifiable as HAdV-D56 or as strains described as having the same penton base, hexon, and/or fiber gene types, and the phylogenetically related genotypes D9, D10, D15, D26, D29, D88, and D94. Collectively, the sixteen HAdV-D56 strains showed a percent identity above 99.9%. The representation in the panel of strains isolated from a diversity of specimens, including conjunctival swabs, urine, vaginal, and neonatal nasopharyngeal swabs, could not identify any specific sequence signatures correlating with any particular tropism. This strongly supports the notion that HAdV-D56, like other members of species HAdV-D, has the ability to infect ocular, respiratory, and genital (both male and female) epithelia.

Our phylogenetic analysis, including complete genomic sequences available in GenBank for viruses with the same penton base, hexon, and fiber gene type identities circulating in the US and Europe in the early 1980s and 1990s provided evidence of the circulation of HAdV-56 prior to 2008. Importantly, the in silico restriction enzyme analysis conducted in this study allowed us to extend our comparisons to published digestion profiles for intermediate strains 15/H9 isolated between 1958 and 1979 in the USA and Europe and characterized antigenically by neutralization and hemagglutination inhibitions assays and by REA by Adrian and colleagues [35,36]. The majority of the D56 strains examined in this study, including the prototype isolated in France in 2008 (D56-VI), showed identical profiles to those reported by these investigators for strains 5399 (Netherlands, 1958), V360 (LA, USA, 1979), and 910 (CA, USA, 1973) [35,36]. Collectively, this body of data establishes important epidemiological connections and strongly suggests that HAdV-D56-like viruses have circulated in various parts of the world since the late 1950s and that HAdV-D56 is not an emergent pathogen as initially thought. Our genomics and proteotyping analysis also predict that, like D94 and D56-I (the only serotyped viruses in the examined panel), all of these viruses with D15/29-like hexon genes and D9-like fiber genes will neutralize as serotype 15 and as intermediate variants 15/H9 if hemagglutination inhibition assays are carried out to contribute information on the fiber antigenic type.

Collectively, these three recent cases of HAdV-D56 neonatal disease contribute to raising awareness of the existing major gaps in knowledge regarding the transmission dynamics resulting in neonatal infection, the interplay between the ocular and genital mucosae during infection by certain members of species HAdV-D, and the need for increased surveillance. The possible sexual transmission of HAdV-D56 and other HAdV-D infections merits further investigation.

Screening of pregnant women for evidence of active genital HAdV infection and documentation of history of household conjunctivitis or acute ocular or respiratory infection during pregnancy may help inform interventions to prevent neonatal infections. Human adenovirus testing must be included in the differential diagnosis of neonatal sepsis

## Figures and Tables

**Figure 1 viruses-13-01105-f001:**
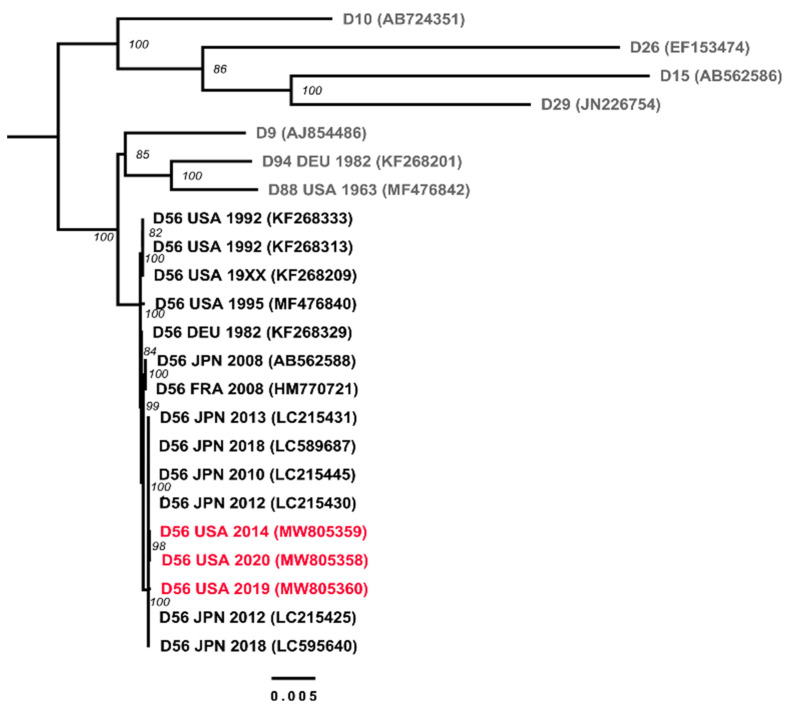
Phylogenetic analysis of the whole genome sequences of USA strains NCH-Pro95/2014, IDR1900044114 and VIR209329 (in red font), previously identified strains of HAdV-D56, and closely related HAdV-D genotypes -D9, -D10, -D15, -D26, -D29, -D88, and -D94. Maximum likelihood phylogenetic tree of types HAdV-D9, -D10, -D15, -D26, -D29, -D56, -D88, and -D94. The bootstrap support is shown next to the branching nodes. The length of the branches represents the number of mutations per nucleotide site according to the scale.

**Figure 2 viruses-13-01105-f002:**
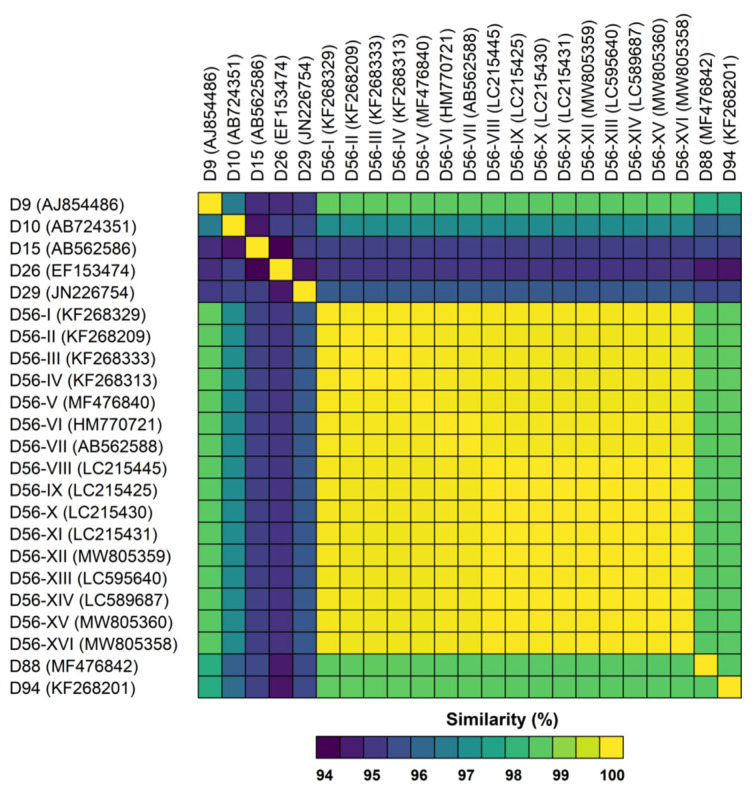
Pairwise analysis of whole genome sequence similarity among genotypes D9, D10, D15, D26, D29, D88, D94, and various strains of the genotype D56. The data matrix is presented as a heat map where the percentage of pairwise sequence similarities are color-coded according to the scale at the bottom of the figure.

**Figure 3 viruses-13-01105-f003:**
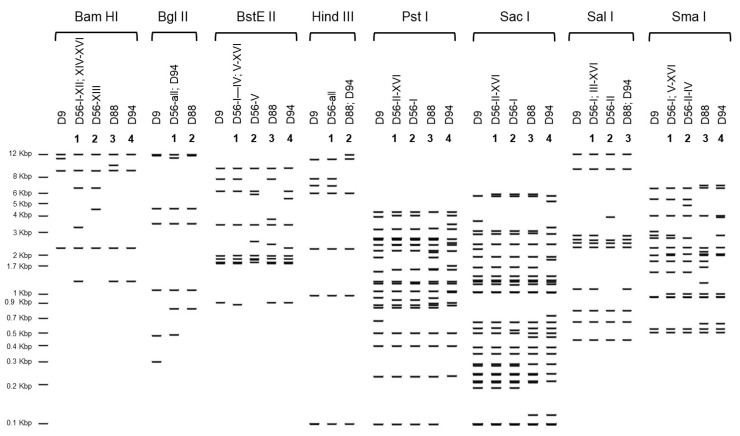
Comparative in silico restriction enzyme analysis of HAdV-D56 strains and closely related genotypes D9, D88, and D94 with endonucleases Bam HI, Bgl II, Bst EII, Hind III, Pst I, Sac I, Sal I, and Sma I. The analysis was run in Geneious Prime using the 1 Kb plus DNA ladder with bands ranging in size between 12 Kbp (at the top) and 100 bp (at the bottom of the scale on the left-hand side of the panel). Numbers 1, 2, 3, and 4 designate distinct restriction profiles for genotypes with H15 and F9.

**Figure 4 viruses-13-01105-f004:**
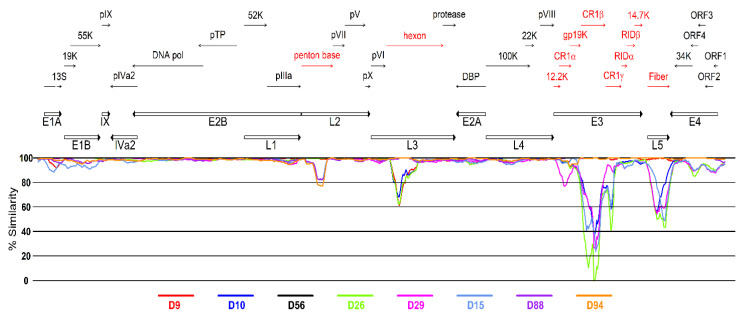
Analysis of genomic sequences for regions of divergence and similarity. The similarity plot was generated using the whole genome sequence comparing the consensus sequence for all HAdV-D56 strains included in the analysis as a query against various selected genotypes (D9, D10, D15, D26, D29, D88, and D94). The plot represents the percent similarity in a 500-nucleotide sliding window and a 50-nucleotide step size with gapped sites removed. Color codes are shown at the bottom of the panel. Regions encoding proteins that were further examined in the proteotyping analysis are colored in red.

**Figure 5 viruses-13-01105-f005:**
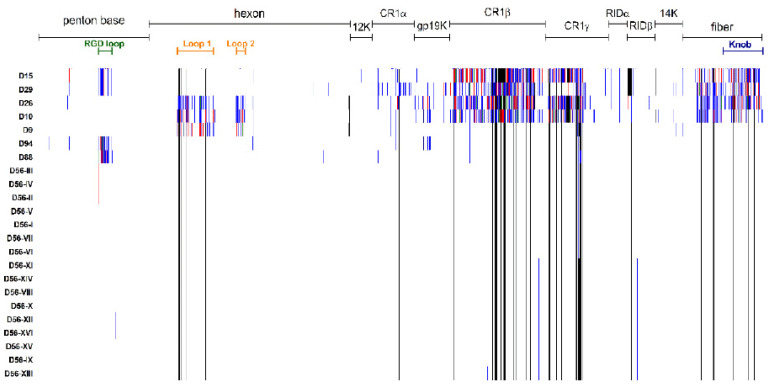
Proteotyping of concatenated selected protein sequences for HAdV-D56 strains I–XVI and closely related HAdV-D genotypes. The vertical axis corresponds to the compared sequences and the horizontal axis corresponds to the amino acid position in the concatenated sequences. The names of the proteins are shown on the top of the figure. The names of the sequences are annotated to the left of the figure. Sites with amino acid polymorphisms are colored accordingly with the frequency across sequences, with blank for the consensus, blue and red for the first and second most frequent polymorphisms for a position; gap sites are colored in black.

**Table 1 viruses-13-01105-t001:** Human mastadenovirus D genotypes and strains included in the genomic analysis.

Strain Designation	Genotype	Assigned ID for This Study	Source	Clinical Illness	Year of Isolation	Place of Isolation	Accession No.	Molecular Type		
								P	H	F
Hicks (p)	D9	D9	Anal specimen	Arthritis	1954	MA, USA	AJ854486	9	9	9
J.J. (p)	D10	D10	Conjunctival swab	Conjunctivitis	195X	D.C., USA	AB724351	10	10	10
CH38 (p)	D15	D15	Conjunctival scrapings	Conjunctivitis	1955	SAU	AB562586	15	15	15
BP-2 (p)	D26	D26	Anal specimen	None	1956	D.C., USA	EF153474	26	26	26
BP-6 (p)	D29	D29	Anal specimen	NA	1959	D.C., USA	JN226754	29	15	29
MEE-MOLD	D88	D88	Respiratory specimen	NA	1963	MA, USA	MF476842	88	15	9
HEIM 00080	D94	D94	NA	NA	1982	DEU	KF268201	33	15	9
HEIM 00081	D56	D56-I	NA	NA	1982	DEU	KF268329	9	15	9
MEEI 00078	D56	D56-II	NA	NA	19XX	MA, USA	KF268209	9	15	9
CL 50	D56	D56-III	NA	NA	1992	PA, USA	KF268333	9	15	9
Pitts 00150	D56	D56-IV	NA	NA	1992	PA, USA	KF268313 ^+++^	9	15	9
MEE-CHBR	D56	D56-V	Vaginal swab	NA	1995	MA, USA	MF476840	9	15	9
p	D56	D56-VI	Lung biopsy	Sepsis ^N^	2008	FRA	HM770721	9	15	9
2307-S	D56	D56-VII	Conjunctival swab	EKC	2008	Sapporo, JPN	AB562588	9	15	9
20101537	D56	D56-VIII	Conjunctival swab	EKC	2010	Kumamoto, JPN	LC215445	9	15	9
20121516	D56	D56-IX	Conjunctival swab	EKC	2012	Kumamoto, JPN	LC215425	9	15	9
20121569	D56	D56-X	Conjunctival swab	EKC	2012	Kumamoto, JPN	LC215430	9	15	9
20131505	D56	D56-XI	Conjunctival swab	EKC	2013	Kumamoto, JPN	LC215431	9	15	9
NCH-Pro95	D56	D56-XII	Respiratory specimen	Sepsis ^N^	2014	OH, USA	**MW805359 ^§^**	9	15	9
TKY3113Asew	D56	D56-XIII	Sewage water	---	2018	Tokyo, JPN	LC595640	9	15	9
TKYAd188507	D56	D56-XIV	Urine	Urethritis	2018	Tokyo, JPN	LC589687	9	15	9
IDR1900044114	D56	D56-XV	Respiratory specimen	Sepsis ^N^	2019	NY, USA	**MW805360 ^§^**	9	15	9
VIR209329	D56	D56-XVI	Respiratory specimen	Sepsis ^N^	2020	PA, USA	**MW805358 ^§^**	9	15	9

p: Prototype strain; NA: information not available; EKC: epidemic keratoconjunctivitis; ^N^ Neonatal sepsis; X: year unknown; **^§^** genomic sequences obtained in this study; P: penton base gene; H: hexon gene; F: fiber gene; ^+^ described in GenBank as (P26H56F56).

**Table 2 viruses-13-01105-t002:** Identified non-synonymous point mutations in the genomes of the US strains of HAdV-D56 (D56-XII, -XV, and -XVI) compared to the prototype strain isolated in France in 2008 (D56-VI).

Nucleotide Position ^§^	Affected Amino Acid Position ^¥^	Gene	Protein Product	D56-VI (HM770721)	D56-XII (MW805359)		D56-XV (MW805360)		D56-XVI (MW805358)	
					AA	Depth ^ჶ^	AA	Depth ^ჶ^	AA	Depth ^ჶ^
788	217	E1a	12S	*L*	**I**	25	**I**	28	**I**	33
1090	519		13S	*M*	*M*	36	**I**	25	*M*	35
1971	398	E1B	19K	*R*	**K**	33	**K**	24	**K**	24
10992	383	L1	52/55K	*S*	**N**	18	**N**	21	**N**	28
14397	913	L2	penton base	*D*	**N**	28	**N**	21	**N**	33
14572	1088		penton base	*P*	**L**	33	*P*	16	**L**	48
15862	197		pV	*H*	**R**	45	**R**	15	**R**	35
28582	1075	E3	CR1β	*E*	**K**	35	**K**	20	**K**	29
28648	1141		CR1β	*I*	**F**	29	**F**	20	**F**	33
29991	140		14.5K	*V*	**A**	29	**A**	15	**A**	39
31837	943	L5	fiber	*D*	**Y**	29	**Y**	16	**Y**	24
33085	346	E4	Orf 4	*S*	**P**	23	**P**	21	**P**	31
33119	20		34K	*S*	**C**	20	S	23	**C**	27
33119	312		Orf 4	*I*	**M**	20	I	23	**M**	27
34405	189		Orf 1	*L*	*L*	28	**F**	21	*L*	28

^§^ Using the genome HM770721 as a reference; ^¥^ In the sequence of the encoded polypeptide; ^ჶ^ Sequence read depth: the number of reads at the specified nucleotide position.

## Data Availability

The genomic sequence data generated for this study are publicly available in GenBank (https://www.ncbi.nlm.nih.gov/genbank) under accession numbers MW805358, MW805359, and MW805360.

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
