# Peer review of "Fatal Neonatal Sepsis Associated with Human Adenovirus Type 56 Infection: Genomic Analysis of Three Recent Cases Detected in the United States"

_viruses, 2021, doi:10.3390/v13061105_

Round 1

Reviewer 1 Report

In this manuscript, Otto et al. present the complete genome sequence for three new epidemiologically unrelated HAdV-D56 isolates.  The authors compare and contrast the complete genome sequence of these new isolates against the prototype strain and other closely related HAdV types. While the finding of 14 non-synonymous mutations between the prototype strain is interesting, the manuscript lacks novel data that would make the identification of these new strains more compelling.

Major Concerns:

  1. Many of the figures focus on the strain divergence between well-characterized HAdVs. For example, the data described in Figure 3 does not present any new data since the divergence is expected based on the known sequence differences in the penton base, hexon, E3, and fiber proteins. 

  1. The new 14 non-synonymous mutations as compared to the prototype and other D56 strains are an interesting finding. However, the authors fail to discuss these mutations in the context of current evolution.  For example, the authors should consider doing molecular clock analysis to examine the evolutionary trajectory of these new isolates.  These data may also provide better context for how long these viruses have been circulating in the human population.

  1. For many of the figures, the authors could provide additional text to understand the rationale for each analysis. In figure 4, little context is given for the proteotyping and the analysis. Further, the conclusions from the analysis are unclear.

Minor Concerns:

  1. The sequencing coverage for each new isolate is unclear. While Illumina likely has high coverage, this information should be added to the manuscript to confirm the non-synonymous mutations.

  1. In Table 2 and Table 3, the top left box should be extended so that “Genotype” and “Nucleotide” are all on one line.

  1. Table 2 Legend (line 216): “various strains og genotype D56”. Should read “of”

Author Response

In this manuscript, Otto et al. present the complete genome sequence for three new epidemiologically unrelated HAdV-D56 isolates.  The authors compare and contrast the complete genome sequence of these new isolates against the prototype strain and other closely related HAdV types. While the finding of 14 non-synonymous mutations between the prototype strain is interesting, the manuscript lacks novel data that would make the identification of these new strains more compelling.

Major Concerns:

  1. Many of the figures focus on the strain divergence between well-characterized HAdVs. For example, the data described in Figure 3 does not present any new data since the divergence is expected based on the known sequence differences in the penton base, hexon, E3, and fiber proteins. 

Response: The point is well taken. However, we still think that to help the reader it is important to present a simple illustration of the areas of the genome where these closely related genotypes display sequence differences so we are opting to keep this figure.

  1. The new 14 non-synonymous mutations as compared to the prototype and other D56 strains are an interesting finding. However, the authors fail to discuss these mutations in the context of current evolution.  For example, the authors should consider doing molecular clock analysis to examine the evolutionary trajectory of these new isolates.  These data may also provide better context for how long these viruses have been circulating in the human population.

Response: This is another point well taken and an interesting path to pursue in future efforts. However, because evolutionary analysis is not the major focus of this paper and because we do not have access to any of the early isolates likely to represent HAdV-D56 variants circulating in 1950s-1960s we do not feel comfortable conducting molecular clock analysis for this paper.

  1. For many of the figures, the authors could provide additional text to understand the rationale for each analysis. In figure 4, little context is given for the proteotyping and the analysis. Further, the conclusions from the analysis are unclear.

Response: We have incorporated some additional text to the legends for Figures 1 and 2.  We have slightly edited the text to provide our rationale for conducting the analysis presented in former Figure 4 (Figures 5 in the revised version of our manuscript) and state our conclusions. Edits are highlighted in yellow on lines 269-271 and 329.

Minor Concerns:

  1. The sequencing coverage for each new isolate is unclear. While Illumina likely has high coverage, this information should be added to the manuscript to confirm the non-synonymous mutations.

Response: We edited the original Table 3 (Table 2 in the revised version) to show the number of reads at each of the positions were non-synonymous point mutations were detected. The table footnote was edited to accommodate this new information. We also edited the results section on lines 201-202 to reported the coverage for each of the sequenced genomes.

  1. In Table 2 and Table 3, the top left box should be extended so that “Genotype” and “Nucleotide” are all on one line.

Response: These issues are the unfortunate result of formatting to fit the Viruses template. They have been corrected. In response to reviewer 2’s suggestions, Table 2 was replaced by a heat map.

  1. Table 2 Legend (line 216): “various strains og genotype D56”. Should read “of”

Response: This typo is corrected in the revised version of the manuscript where the original Table 2 is replaced by a new figure (new Figure 2) displaying the percent pairwise sequence similarities among the examined strains using a heat map.

Reviewer 2 Report

The manuscript of Otto et al. (Kajon) provides new insight into an uncommon human adenovirus (HAdV) that was first identified in a case of fatal neonatal sepsis. This species D virus, which was isolated in 2008 and classified in 2011 as HAdV-D56, has been identified in three more fatal cases of neonatal sepsis in the United States. A more comprehensive analysis of published sequences suggest that virtually identical viruses have circulated in the human population around the world since the 1950s.

This report provides new insight on an important human pathogen that was mistakenly thought to have recently emerged into the population. The molecular analysis of the virus is not sufficient to provide any insight into its pathogenesis or spread but provides a very strong basis for pursuing these important lines of investigation. The report is exceptionally well-written with a very clear description of the methodology and the sequence analysis.

I can offer only one suggestion to improve the clarity of this report. Table 2 appears to be intended to convey the dramatic similarity of the various HAdV-D56 isolates. However, the preponderance of similar digits in a very font size makes this a rather ineffective form of communication. Perhaps this table could be replaced with visualization such as a heat map that more effectively displays the same information.

Author Response

The manuscript of Otto et al. (Kajon) provides new insight into an uncommon human adenovirus (HAdV) that was first identified in a case of fatal neonatal sepsis. This species D virus, which was isolated in 2008 and classified in 2011 as HAdV-D56, has been identified in three more fatal cases of neonatal sepsis in the United States. A more comprehensive analysis of published sequences suggest that virtually identical viruses have circulated in the human population around the world since the 1950s.

This report provides new insight on an important human pathogen that was mistakenly thought to have recently emerged into the population. The molecular analysis of the virus is not sufficient to provide any insight into its pathogenesis or spread but provides a very strong basis for pursuing these important lines of investigation. The report is exceptionally well-written with a very clear description of the methodology and the sequence analysis.

I can offer only one suggestion to improve the clarity of this report. Table 2 appears to be intended to convey the dramatic similarity of the various HAdV-D56 isolates. However, the preponderance of similar digits in a very font size makes this a rather ineffective form of communication. Perhaps this table could be replaced with visualization such as a heat map that more effectively displays the same information.

Response: We appreciate this suggestion. We actually think it is a great idea in view of the space constraints to present table 2. We have replaced Table 2 with a new Figure 2, displaying the sequence similarity comparisons using a heat map. The text in the Methods section of the paper was edited to read “The similarity between pairs of compared sequences was expressed as a percentage and a heat map graphical representation of the data matrix was built using R [22]” on lines 174-176. A new reference #22, Fox et al, 2005, was added to the reference list.

Reviewer 3 Report

Some types of adenovirus are associated with a broad range of severe clinical manifestations in respiratory, lymphoid, gastrointestinal, and ocular tissues, and even cause death. In this study, the authors report three new strains of HAdV-56 isolated from epidemiologically unrelated fatal cases of neonatal sepsis detected in the United States between 2014 and 2020. The analyses indicated that their genome sequences were 99.9% identical to one another and to the prototype strain isolated in 2008 from a similar context of disease. The phylogenetic reconstruction revealed a highly supported clustering of all HAdV-56 strains isolated in various countries since 1982. Therefore, the risk of severe prenatal or perinatal HAdV-D56 infection must be considered.

The manuscript was well organized, and well written, and can be accepted.

Minor error:

Line 146: “are available under accession numbers MW805358, MW805358 and MW805358”.

The above accession numbers should be different.

Author Response

Some types of adenovirus are associated with a broad range of severe clinical manifestations in respiratory, lymphoid, gastrointestinal, and ocular tissues, and even cause death. In this study, the authors report three new strains of HAdV-56 isolated from epidemiologically unrelated fatal cases of neonatal sepsis detected in the United States between 2014 and 2020. The analyses indicated that their genome sequences were 99.9% identical to one another and to the prototype strain isolated in 2008 from a similar context of disease. The phylogenetic reconstruction revealed a highly supported clustering of all HAdV-56 strains isolated in various countries since 1982. Therefore, the risk of severe prenatal or perinatal HAdV-D56 infection must be considered.

The manuscript was well organized, and well written, and can be accepted.

Minor error:

Line 146: “are available under accession numbers MW805358, MW805358 and MW805358”.

The above accession numbers should be different.

Response: We appreciate the positive feedback from this reviewer. The accession numbers listed on line 146 in the original version of the manuscript (lines 149-150 in this revised version) are now corrected. In our submission to GenBank, we requested release of the sequences for public access upon publication of the paper. We apologize for any confusion this may have caused.

Round 2

Reviewer 1 Report

No additional comments or suggestions.